# Reconstruction of evolving nanostructures in ultrathin films with X-ray waveguide fluorescence holography

Zhang Jiang [1]✉, Joseph W. Strzalka [1], Donald A. Walko[1] & Jin Wang[1]✉

Controlled synthesis of nanostructure ultrathin films is critical for applications in nanoelectronics, photonics, and energy generation and storage. The paucity of structural probes that are sensitive to nanometer-thick films and also capable of in-operando conditions with high spatiotemporal resolutions limits the understanding of morphology and dynamics in ultrathin films. Similar to X-ray fluorescence holography for crystals, where holograms are formed through the interference between the reference and the object waves, we demonstrated that an ultrathin film, being an X-ray waveguide, can also generate fluorescence holograms as a result of the establishment of X-ray standing waves. Coupled with model-independent reconstruction algorithms based on rigorous dynamical scattering theories, the thin-film-based X-ray waveguide fluorescence holography becomes a unique in situ and time-resolved imaging probe capable of elucidating the real-time nanostructure kinetics with unprecedented resolutions. Combined with chemical sensitive spectroscopic analysis, the reconstruction can yield element-specific morphology of embedding nanostructures in ultrathin films.

[1] Advanced Photon Source, Argonne National Laboratory, Lemont, IL 60439, USA. ✉email: zjiang@anl.gov; wangj@aps.anl.gov

luorescence is the emission of light or radiation by certain substances as a result of absorbing incident radiation of a shorter wavelength or higher photon energy. Applications of fluorescence such as in spectroscopy and microscopy do not utilize its steradian sensitivity because the directly emitted fluorescence is an isotropic outgoing spherical wave. However, an anisotropic intensity distribution of the fluorescence can be induced when the fluorescence is modulated by local environmental inhomogeneities near its emitting source due to interference of the fluorescence waves. This concept has been explored in the X-ray regime as the X-ray fluorescence holography (XFH)[1–3] for crystalline samples, where local atomic structures can be reconstructed from fluorescence holograms with subatomic spatial resolution. Figure 1a schematically shows the normal mode XFH where fluorescence from an emitter atom and that scattered off from an object atom interfere to form a spatially varying interference pattern in the far-field, a.k.a. fluorescence hologram. In contrast, XFH in the inverse mode is done by scanning the incident angle while recording the fluorescence signal that varies due to the interference between the incoming reference wave and the object wave at the position of the emitter atom (Fig. 1b). While XFH in ordered crystals is three dimensional, we speculate that XFH can be generated to display a lower-dimensional intensity distribution in thin films consisting of layered nanostructures that are confined in the direction normal to the film surface. This is because the film acts as an electromagnetic waveguide so that the reflection at the interfaces confines the waves to the interior of the waveguide and these waves interfere constructively to redistribute the electric field intensity (EFI) normal to the waveguide[5–10]. Hence, the originally isotropic fluorescence wave is modulated within the waveguide, creating a concentric cone-shaped hologram when it leaves the waveguide (Fig. 1c). A similar effect has been observed in the crystallography of single crystals containing fluorescence atoms[11,12] or their Kossel diffractions with a divergent beam[13].

In this work, we illustrate the principle of XFH for a thin-film waveguide and demonstrate that when applied to a film consisting of fluorescence substances, it becomes an in situ and time-resolved imaging technique with sub-nanometer spatial resolution—X-ray waveguide fluorescence holography (XWFH)—for embedded nanostructures and their kinetics in the film. In conventional XFH (normal and inverse modes), holographic reconstruction is achieved via a back-propagation of the far-field hologram in the framework of kinematic approximation[14–18], and the non-kinematic effects such as mode mixing, self-interference, multiple scattering and extinction often unavoidable in actual experimental conditions cause problems such as ghost patterns or false atomic images. In contrast, XWFH takes advantage of the inherent dynamical effects in a waveguide and requires the reconstruction to be performed in the framework of the dynamical theory. For that purpose, a rigorous dynamical theory for thin films has been developed. A model-independent reconstruction algorithm has also been designed based on Bayesian interference. This algorithm reconstructs the coefficients of the cubic b-spline basis for a depth profile with an efficient Markov Chain Monte Carlo sampler based on the concept analogous to the Hamiltonian dynamics in classical mechanics. Applying this algorithm to a mixed-mode XWFH carried out at both grazing-incidence and exit angles, we can take many advantages of the dynamical scattering effects and treat these effects as additional

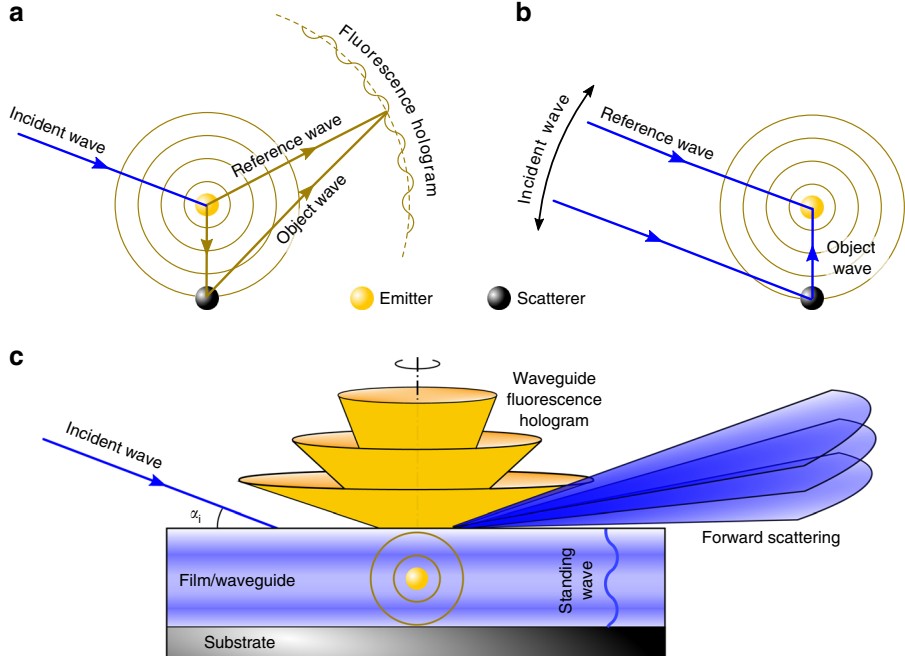

**Fig. 1 Schematics of X-ray fluorescence holography (XFH) operation modes. a** In the normal XFH mode[1], an emitting atom (i.e. emitter) gives out a spherically outgoing X-ray fluorescence wave. An object wave is formed when this outgoing wave is scattered by a nearby atom (i.e. scatterer) and then interferes with the unperturbed outgoing reference wave, producing a spatially distributed hologram. Lines in blue and gold colors represent waves of elastic and fluorescence energies, respectively. **b** In the inverse XFH mode[4], the exciting wave at the emitter is a result of interference of the unperturbed incident wave and the scattered incident wave from the scatterer. The hologram is constructed by scanning the incident wave and recording the integrated fluorescence intensity at a fixed detector position. **c** In the mixed-mode X-ray waveguide fluorescence holography (XWFH) at grazing-incidence and exit angles, X-ray standing waves are created within the waveguide for both the elastic and fluorescence energies. The intensity of fluorescence leaving the waveguide concentrates at discrete exit angles, producing a concentric cone-like hologram whose axis of rotation is perpendicular to the film surface. The forward elastic scattering is also modulated by the waveguide and its angular dependence is measured via grazing-incidence small-angle X-ray scattering (GISAXS).

constraints for better and fast convergence and reconstruction qualities. To illustrate these concepts, we selected gold nano-particle monolayers embedded within supported and capping polymer films, i.e., sandwiching layers, as model systems. Combinations of two molecular weights (low and high) for the sandwiching polymer layers were chosen in order for the in situ study of the diffusion kinetics of the nanoparticles upon thermal annealing. Buried nanostructures in thin films have often been measured with forward scattering-based techniques such as grazing-incidence small-angle X-ray scattering (GISAXS) and reflectivity. With XWFH, we were able to monitor the diffusion kinetics of the nanoparticles because the broadened nanoparticle distribution upon thermal annealing alters the waveguide conditions which were detected as the variation of the angular dependence of the gold fluorescence hologram. The advantages of performing XWFH on nanostructured thin films emerged when the dynamically reconstructed gold atomic number density distribution was compared to the result from the reflectivity and the structures from the model fitting of the simultaneously collected GISAXS.

## Results

**Principle of XWFH.** We first need to understand the intensity distribution of the electric field in the waveguide as it directly relates to the yield and the spatial distribution of fluorescence signals. With incident energy of 12.11 keV (denoted as elastic energy or excitation energy below), the gold's $L_3 2p_{3/2}$ (11.919 keV) energy level is excited, with corresponding dominant X-ray emission energies $L\alpha_1$ (9.713 keV), $L\alpha_2$ (9.628 keV), $L\beta_2$ (11.585 keV), and $L\beta_{15}$ (11.566 keV). However, the subtle split between the two $L\alpha$ lines, as well as the two $L\beta$ lines, is indistinguishable given the energy and angular resolution of the detector and the instrumentation setup. Therefore, they are treated as two rather than four emission lines: $L\alpha_{(1,2)}$ and $L\beta_{(2,15)}$, whose relative fluorescence yields (fluorescence intensity from an atom) are fixed at tabulated values[19] during the reconstruction.

Figure 2a shows the calculated EFI as a function of depth and incident angle (or exit angle by optical reciprocity theorem) for the elastic energy. The calculation is done using reconstructed parameters (see text in the next three sub-sections and Fig. 3c) for an as-cast LH sample (nanoparticle monolayer sandwiched between a supporting layer of low molecular weight and a capping layer of high molecular weight; see Methods for Materials and samples). Near-surface evanescent wave[20] is created if the incident angle is less than the critical angle for total-external reflection of the polymer film ($\alpha_{c,flim}|_e = 0.102°$). If the incident angle is above $\alpha_{c,flim}|_e$, the electric field penetrates into the film and gets amplified at certain depths when the incident angle coincides with special angles between $\alpha_{c,flim}|_e$ and $\alpha_{c,Pd}|_e$ (the critical angle of the supporting Pd mirror which is 0.317° for the incident energy). This EFI enhancement is known as the waveguide effect or X-ray standing wave (XSW) in grazing-angle conditions[5–9], and these special angles are denoted as anti-node angles. However, the EFI enhancement quickly diminishes beyond $\alpha_{c,Pd}|_e$, as a consequence of loss due to X-ray penetration into the substrate. Therefore, if one is to utilize this enhancement for a better signal-to-noise ratio, the incident angle should be kept below the critical angle of the substrate. For example, an incident angle of 0.125° is used in this study for both XWFH and GISAXS.

By the optical reciprocity theorem, the same EFI enhancement effect can be also observed at the exit angle side for internally excited fluorescence. The critical angles of the polymer film are $\alpha_{c,flim}|_{L\alpha} = 0.127°$ and $\alpha_{c,flim}|_{L\beta} = 0.107°$ for $L\alpha_{(1,2)}$ and $L\beta_{(2,15)}$, respectively. If the exit angle is below 0.125°, for example, only the evanescent wave exists for $L\alpha_{(1,2)}$ fluorescence. Both the elastically

scattered intensity and the fluorescence intensity are proportional to the number of gold atoms as well as the magnitudes of the incident electric fields these atoms are exposed to. In other words, although the independently established EFI at each relevant energy (one elastic and two fluorescence energies) has its own angle (Fig. 2b) and depth (Fig. 2c) dependence, and contributes incoherently to the total XWFH (Fig. 3c), they are interrelated through the same electron density profile $\rho(z)$ of the waveguide. Therefore, the key to reconstructing the angle-resolved XWFH holograms is to determine the common dependence of these EFIs on $\rho(z)$. This needs to be done iteratively with are construction based on dynamical scattering theory described as follows.

**Dynamical theory for reconstructing the mixed-mode XWFH holograms.** The total intensity measured on the fluorescence pixel-array detector consists of fluorescence intensity $I_f(\alpha_i, \alpha_f)$ and elastic scattering background $I_e(\alpha_i, \alpha_f)$,

$$I_F(\alpha_i, \alpha_f) = wI_f(\alpha_i, \alpha_f) + (1 - w)I_e(\alpha_i, \alpha_f), \quad (1)$$

where $\alpha_i$ and $\alpha_f$ are respectively the incident and exit angles with respect to the film surface, and $w$ is a weight factor for the fluorescence contribution.

The emission power of the immediate fluorescence of a gold atom is proportional to the square of the exciting incident electric field $E(z, \alpha_i, \lambda_e, \rho(z))$. This incident field varies at different depths of a waveguide and depends on the angle $\alpha_i$ and wavelength $\lambda_e$ of the incident wave, as well as the overall electron density profile $\rho(z)$ of the waveguide. On the emission side, the waveguide modulates the fluorescence and establishes a fluorescence electric field $E(z, \alpha_f, \lambda_f, \rho(z))$, where $\alpha_f$ and $\lambda_f$ are the exit angle and wavelength of the fluorescence wave, respectively. In the presence of strong multiple reflections at the waveguide interfaces at grazing angles, these electric fields need to be computed with the dynamical theory. For one-dimensional scenarios such as in the normal direction of the waveguide, Parratt's recursive method is often adopted for the electric field computation (see Methods for Electric field computation).

Summing over the emission spectrum, a.k.a. relative fluorescence yield $Y_{Au}(\lambda_f)$, we can write the fluorescence intensity as

$$I_f(\alpha_i, \alpha_f) \propto \int d\lambda_f Y_{Au}(\lambda_f)$$
$$\times \int dz |E(z, \alpha_i, \lambda_e, \rho(z))|^2 \phi_{Au}(z) \left| E(z, \alpha_f, \lambda_f, \rho(z)) \right|^2, \quad (2)$$

where $\phi_{Au}(z)$ the gold atomic number density is the ultimate goal of the reconstruction. It is related to $\rho(z)$ and can be recursively reconstructed (see Sub-section XWFH reconstruction algorithm).

Similarly, the elastic background is written as

$$I_e(\alpha_i, \alpha_f) \propto \int dz |E(z, \alpha_i, \lambda_e, \rho(z))|^2 \sum_j \sigma_j \phi_j(z) \left| E(z, \alpha_f, \lambda_e, \rho(z)) \right|^2, \quad (3)$$

where $\phi_j(z)$ is the atomic number density for the $j$th element, and $\sigma_j$ is its elastic scattering cross-section[19]. The summation goes over every element in the entire sample: Si, Cr, Pd, Au, as well as C, H, and O in the polymers.

The relation of the mixed-mode XWFH at both grazing-incidence and exit angles to the conventional XFH can be understood as follows. In the perspective of the normal XFH, $E(z, \alpha_i, \lambda_e, \rho(z))$ is the incident wave, while $E(z, \alpha_f, \lambda_f, \rho(z))$ is the hologram as a result of the self-interference of fluorescence within the waveguide. On the other hand, the excitation field $E(z, \alpha_i, \lambda_e, \rho(z))$ can be also seen, from the inverse XFH's point

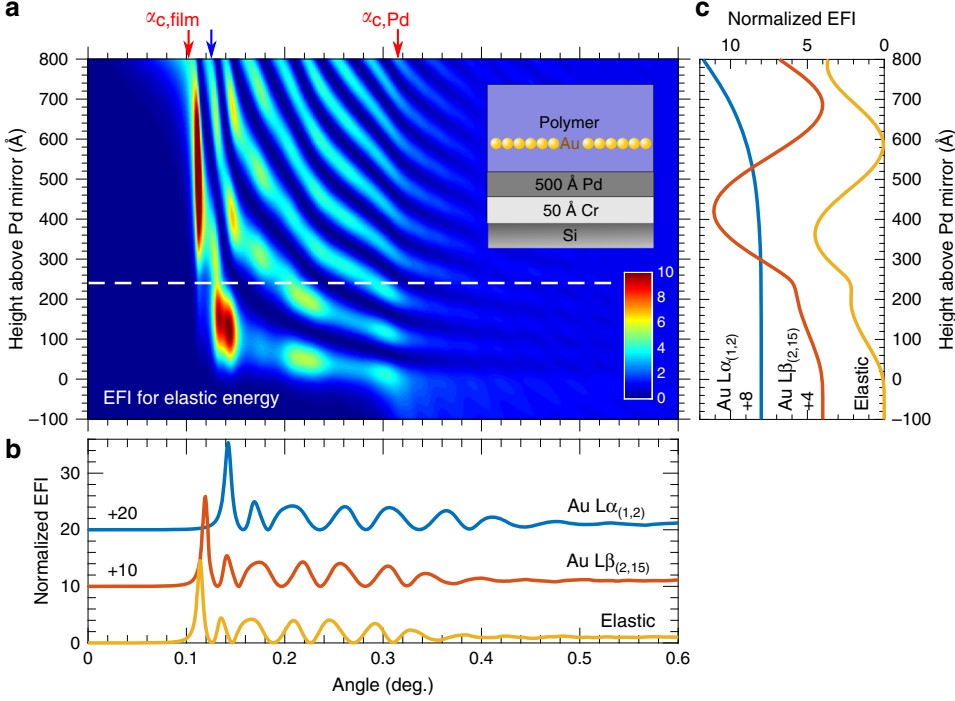

**Fig. 2 Electric field intensity (EFI) distribution. a** EFI for the elastic energy (12.11 keV) as a function of grazing angle and height above the Pd mirror for the LH sample whose the gold monolayer is ~250 Å above the mirror (white dashed line). Parameters for the calculation are from the reconstructed XWFH before the thermal annealing (details described in the text). Two red arrows indicate the critical angles for total-external reflection of the polymer film ($\alpha_{c,flim}|_e = 0.102°$) and the Pd mirror ($\alpha_{c,Pd}|_e = 0.317°$), respectively, at the elastic energy. Inset illustrates the layers of the sandwiched film. **b** Line profiles of the EFI at the gold monolayer height for three energies: elastic, gold L$\alpha_{(1,2)}$, and L$\beta_{(2,15)}$. **c** Line profiles of the normalized EFI at a grazing angle of 0.125° (indicated by the blue arrow in **a**) for the three energies.

of view, as the consequence of the interference of the gold atoms and the entire waveguide. Unlike in the conventional XFH, it is impossible to distinguish the reference from object waves here, because (1) the strong multiple reflections occur at the interfaces for angles in the vicinity of the total-external reflection; and (2) gold is not only the emitter but also the scatterer in the waveguide and has a significant effect on the incident electric field that subsequently determines the total fluorescence intensity. These intertwining effects cannot be handled with conventional reconstruction methods based on the kinematical approximation. Instead, they can only be approached with the dynamical formula as described above.

**XWFH reconstruction algorithm.** Given prior information about the gold atomic number density profile $\phi_{Au}(z)$, one may model the profile approximately with appropriate empirical analytical functions. For example, Gaussian gives a reasonably stratifying description of the sandwiched gold monolayer in the present study, where the number density profile is determined by three parameters: mean height, root-mean-squared width, and the total amount of gold atoms. These parameters can be determined by $\chi^2$-minimization or other optimization methods. However, in order to develop a generalized reconstruction method for any arbitrary profiles, we introduce a Bayesian-inference based model-independent algorithm, where no prior knowledge and profile modeling is required. In this approach, an arbitrary smooth profile can be numerically represented by cubic b-splines. Specifically, $\phi_{Au}(z)$ can be written as a linear combination of a set of $N$ cubic b-spline basis $\{B_i(z)\}$ whose first and last basis end with zero at the film/substrate and film/helium interfaces (i.e. neither do gold atoms leave the film nor penetrate into

the substrate)[21–23] (also see Supplementary Note 1),

$$\phi_{Au}(z) = N_{Au} \sum_{i=1}^{N} C_i B_i(z),$$

$$\text{subject to } C_i \geq 0 \text{ and } \int_{-\infty}^{\infty} dz \sum_{i=1}^{N} C_i B_i(z) = 1, \quad (4)$$

where $N_{Au}$ is the total number of gold atoms and its value corresponds to the nominal thickness during the thermal deposition. $C_i$ are b-spline coefficients to be reconstructed. In order for parameter parsimony and overall curve smoothness, regularizations on $C_i$ and their variations $\Delta C_i$ are applied. Here we used a modified version of the fused lasso regularization[24] for the cost function such that

$$J = \sum_{j=1}^{M} \left( I_{F,j}^{mea} - I_{F,j}^{cal} \right)^2 + \beta_1 \sum_{i=1}^{N} |C_i| + \beta_2 \sum_{i=1}^{N-1} |\Delta C_i|^2, \quad (5)$$

where the 1st term is sum-square-residual (SSR) of the $M$ experiment data points, the 2nd term encourages the sparsity of the $N$ basis splines, and the 3rd term encourages the overall smoothness. $\beta_{1,2}$ are corresponding penalty parameters. In addition, mean-square-residual method (MSR) was used to determine the minimal total number of basis splines (insets of Fig. 3c, h).

Although many nonlinear optimization solvers can be used to minimize this regularized cost function, the reconstruction is not necessarily guaranteed to converge or the convergence may consume extraordinarily long time due to the high-dimensional parameter space (e.g., 30 spline coefficients in Fig. 3c). Hence we adopted an efficient Bayesian-inference method, the recently developed Hamiltonian Markov Chain Monte Carlo (HMCMC)

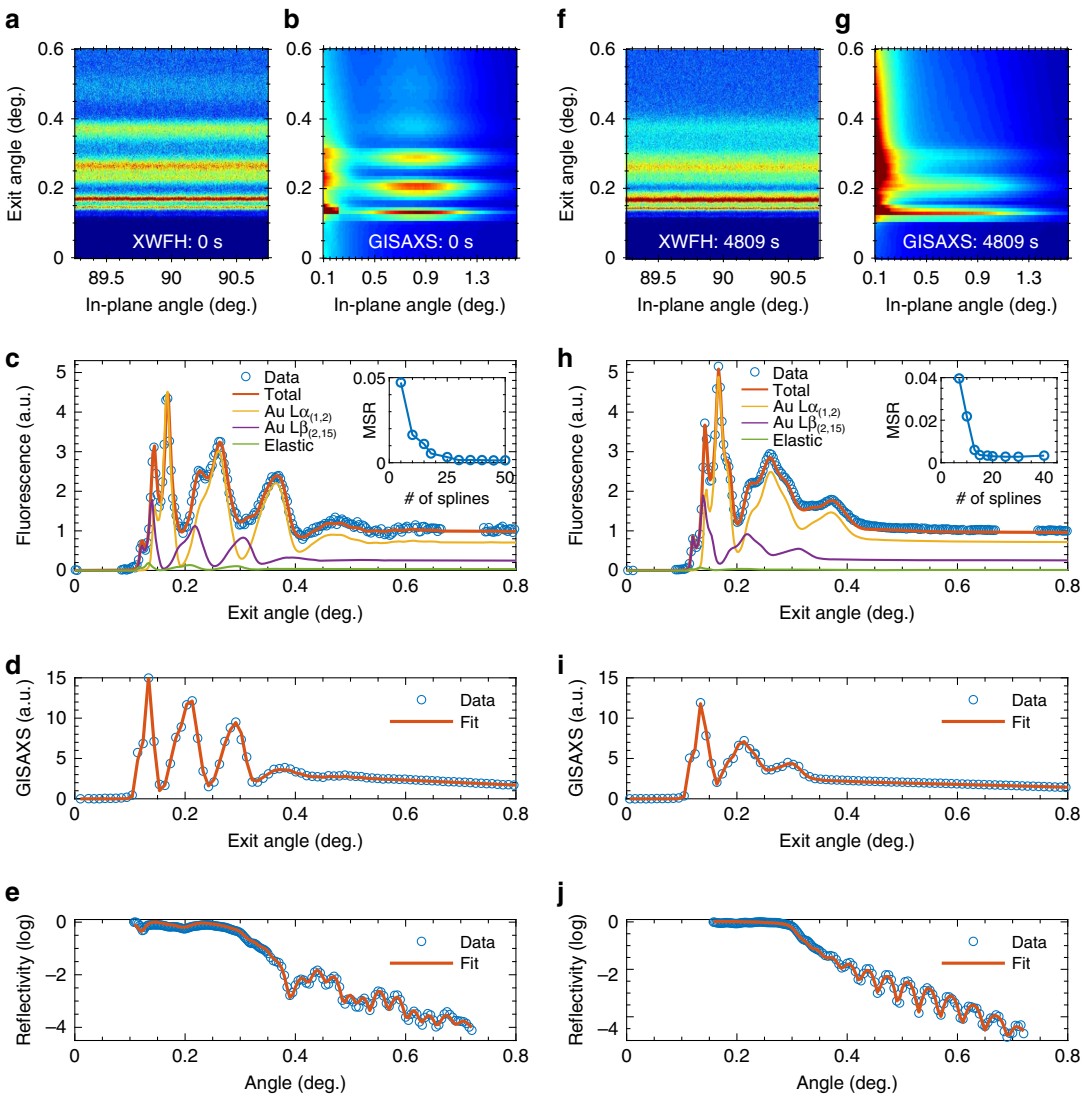

**Fig. 3 XWFH, GISAXS, and reflectivity measured in the beginning and at the end of the thermal annealing for the LH sample.** (**a**) and (**b**) are XWFH and GISAXS patterns taken at an incident angle of 0.125°. GISAXS pattern is mirror-symmetrical with respect to the forward scattering direction (i.e. 0° in-plane angle), so only the right half is shown in (**b**). **c** In-plane integrated experimental and reconstructed XWFH data. Individual contributions from the Lα$_{(1,2)}$, Lβ$_{(2,15)}$ emissions and the elastic background are displayed. The inset MSR (mean-square residual) analysis indicates that 30 cubic b-splines is sufficient to describe the gold atomic number density profile. (**d**) and (**e**) are, respectively, GISAXS and reflectivity data with the best fit. (**f–j**) are the result at the end (4809 s) of the thermal annealing. The inset in (**h**) suggests 15 cubic b-splines.

sampling method[25,26], to estimate these parameters as well as the reconstruction confidence (see Supplementary Note 2). This method, bearing some analogy to the concept of Hamiltonian dynamics in classical mechanics, explores the parameter space more efficiently for high-dimensional problems than conventional optimization algorithms. Assuming that the probability of the measured intensity at the $j$th data point is a normal distribution such that $I_{F,j}^{mea} \sim \mathcal{N}\left(I_{F,j}^{cal}, \sigma^2\right)$, the cost function $J$ can be transformed to the potential energy in the language of HMCMC,

$$U = \frac{1}{2\sigma^2} \sum_{j=1}^{M} \left(I_{F,j}^{mea} - I_{F,j}^{cal}\right)^2 + M \log \sqrt{2\pi\sigma^2}$$
$$+ \frac{1}{2\sigma^2} \left(\beta_1 \sum_{i=1}^{N} |C_i| + \beta_2 \sum_{i=1}^{N-1} |\Delta C_i|^2\right).$$
(6)

Starting with a randomized or a guessed position in the parameter space, HMCMC stochastically explores the parameter space to

generate a sequence of parameter samples. After the chain becomes stationary (named burn-in or warm-up), the sequence can be used for the inferences on the parameters as well as the gold atomic number density profile and its confidence intervals. HMCMC often converges quickly to the target probability distribution because the ergodic property of HMCMC algorithms avoids local traps in some subsets of the parameter space. For example, even starting with the weakest guess (i.e. gold atoms are evenly distributed throughout the entire polymer film), the convergence to the unique solution appears in only a few iterations and becomes stable thereafter (see Supplementary Note 2).

**Comparison of XWFH to GISAXS and reflectivity.** Typical XWFH hologram is displayed in Fig. 3a for the sandwiched LH sample before thermal annealing (see Methods for Materials and samples). It was taken with a pixel-array detector mounted at a 90° in-plane angle (see Methods for Experimental details). The propagation of the fluorescence from point-like emitting atoms is

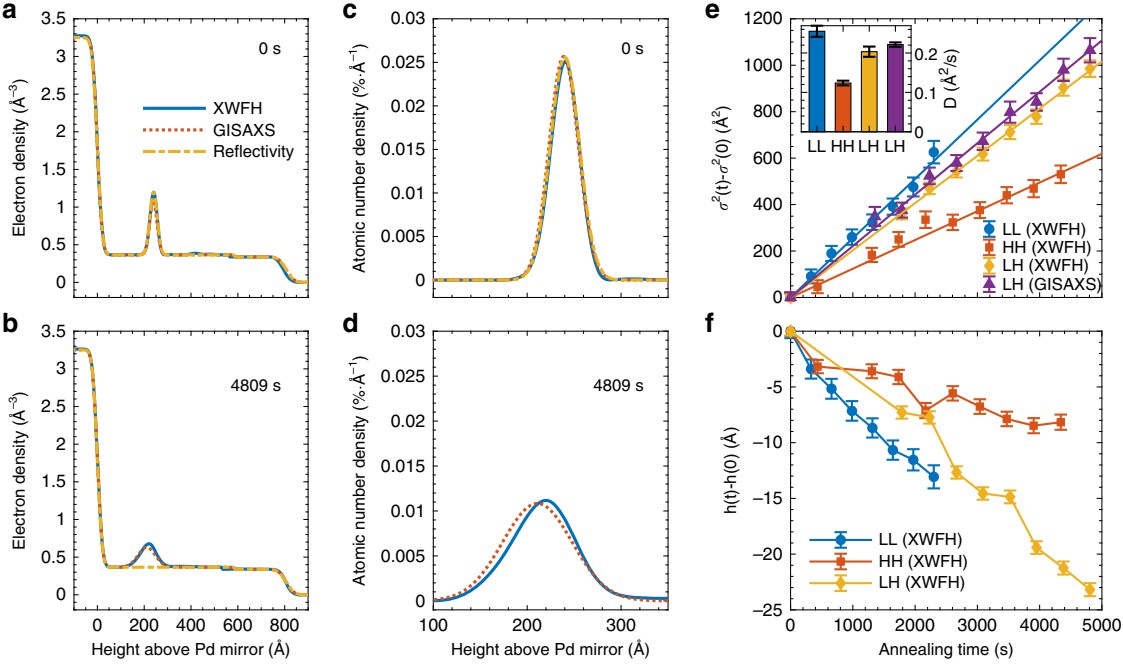

**Fig. 4 Result of XWFH reconstruction compared to GISAXS and reflectivity results. a, b** Electron density profiles before and after thermal annealing. The high-density region on the left side and the zero-density on the right side represent Pd mirror and helium, respectively. (**c**) and (**d**) correspond to the normalized gold atomic number density distributions. The result of reflectivity is not shown in (**d**) as the reflectivity loses the sensitivity for small density variations. **e** Change of variance $\sigma^2(t) - \sigma^2(0)$ due to out-of-plane diffusion of the gold monolayer in LL, HH, and LH samples. Error bars represent the uncertainties. The result of the GISAXS on LH is also shown to compare with the XWFH reconstruction. Solid lines are best fit to the diffusion equation (described in the text) with the fitted out-of-plane diffusion coefficient shown in the inset. (**f**) Mean height change of the gold monolayer with the error bars for uncertainties. The solid lines connecting the data points are a guide to the eye. Source data for (**e**) and (**f**) are provided in Supplementary Tables 1–3.

modulated in the normal direction of the waveguide, leading to a concentric cone-like spatial power distribution when the fluorescence leaves waveguide. Since this distribution is isotropic in the plane of the waveguide and depends only on the exit angle, we can integrate it over a wide range of in-plane angles to give a better signal-to-noise ratio in the one-dimensional XWFH (Fig. 3c). XWFH is then reconstructed using Eqs. (1)–(3) with respect to the number distribution profile of the gold atoms $\phi_{Au}(z)$ which is model-independently represented by cubic b-splines.

In contrast, GISAXS is a conventional surface scattering technique to measure nanostructures in thin films[27]. It is performed in the forward direction at low in-plane and exit angles, so its elastic scattering cross-section in the forward direction is many orders of magnitude stronger than the fluorescence[19]. Figure 3b shows the GISAXS pattern recorded on another pixel-array detector mounted in the forward direction. It was taken simultaneously with the XWFH. Unlike XWFH whose reconstruction does not require any morphological and structural assumptions about the nanoparticles, the intensity distribution of GISAXS depends on the size and shape distribution of the nanoparticles as well as their positional correlations. Therefore, GISAXS needs to be fitted rather than reconstructed with the assistance of modeling in the framework of the distorted wave Born approximation (see Methods for Theory of GISAXS analysis). The best fit result (Fig. 3d) is obtained with the following modeling assumptions: the nanoparticles are spheres; they form a two-dimensional monolayer superlattice whose nanoparticle number density in the film normal direction is modeled as a Gaussian, and the inter-particle correlation in the plane follows a one-dimensional paracrystal model[28].

As a third independent surface technique, reflectivity was recorded in the forward scattering direction but with both incident and exit angle scanned (thus losing time-resolution) at identical values (see Methods for Reflectivity analysis). These three independent techniques yield very similar electron density profile (Fig. 4a), or equivalently the atomic number density (Fig. 4c). It is noticed that the $\phi_{Au}(z)$ obtained from GISAXS is slightly broader than those of the XWFH and reflectivity, which is attributed to the insufficient exit angle sampling (due to limited sample-detector distance) in the GISAXS setup and GISAXS's tendency of slightly overestimating the distribution variance (see Supplementary Note 3).

The diffusion of the gold nanoparticle monolayer upon thermal annealing alters the nanoparticle number density $\phi_{np}(z)$ in the film normal, hence the atomic number density $\phi_{Au}(z)$ and the overall electron density profile. As the width of the initially well-defined gold monolayer gets broader, the overall density evolves from a sandwiched film with well-defined film/gold interfaces towards a more mixed film with obscure interfaces. This smears out high-order modes of both the incident standing wave (for the elastic energy) that excites the gold atoms and the fluorescence hologram cones. Meanwhile, the amplitude contrast of the EFI oscillations reduces, as displayed in Fig. 3f and its integrated line curve in Fig. 3h. Identical phenomena are also observed in Fig. 3g, i for GISAXS. For GISAXS, there is an additional dependence on the inter-particle correlation exhibited as a tendency of in-plane aggregation upon thermal annealing (Fig. 3g)[29,30]. Both XWFH reconstruction and GISAXS fitting give similar results as for the electron density profile showing much smaller density variations across the film as the nanoparticle monolayer diffuses out in the normal direction upon thermal annealing. However, the reflectivity completely loses its sensitivity to these small density

variations, because a more parsimonious single-layer model with an averaged uniform electron density fits the reflectivity sufficiently well (Fig. 4b). This can be ascribed to the dependence of the reflectivity sensitivity not only on the highest measured angle (or wave vector transfer) but more importantly on the largest density contrast throughout the layer stacking in this case. To be specific, the density variation of the gold–polymer composite mixture is overwhelmed by the much higher contrast of helium/Pd that dominates the reflectivity sensitivity. In contrast, the sensitivity of XWFH and GISAXS arises from the modulation of the electric field in the waveguide, rather than solely the density contrast between layers. In other words, the perturbation of the gold to the depth-dependent incident electric field, as well as its angularly modulated fluorescence (in XWFH) and elastic scattering (in GISAXS) due to subsequent exposure to this incident electric field, provides much higher sensitivity than reflectivity.

**Diffusion kinetics of gold nanoparticle monolayer**. By monitoring their kinetics or dynamics in a polymer melt, nanoparticles can be used as markers to determine rheological properties such as viscosity of a polymer matrix. In this work, the time dependence of the gold nanoparticle distribution is used to determine the out-of-plane (normal to the film surface) diffusion coefficient in the polymer film. Figure 4e displays the results of the variance of the $\phi_{Au}(z)$ profile with respect to the beginning of the annealing. The diffusion coefficient $D$ is obtained via $\sigma^2(t) - \sigma^2(0) = Dt$. As pointed previously, GISAXS inclines to return a broader distribution than XWFH as seen for the LH sample, thus GISAXS yields a slightly larger diffusion coefficient. The variance plot clearly indicates the trend of the diffusion coefficient on the molecular weights of the two sandwiching layers. Sample with both supporting and capping layers made of low (high) molecular weight polymer exhibits the fastest (slowest) diffusion kinetics, while the sample with layers made of polymers of mixed molecular weight falls in between. It is also noticed that the mean height of the distribution of three samples slightly moves towards the Pd substrate (Fig. 4f), indicating the van der Walls attraction between the heavy metals (Au and Pd). As expected, this overall translational movement towards the substrate is the least for the HH sample with both layers made of high molecular weight polymers.

## Discussion

The mixed-mode X-ray waveguide fluorescence can be simplified and performed at either grazing-incidence or grazing-exit geometries. In the former configuration, a detector of a very large solid angle is desired in order to collect as much fluorescence as possible for a better signal-noise ratio. The hologram is constructed by scanning the incidence angle of a well collimated monochromatic X-ray beam as is done in the conventional inverse XFH and the similar scenario of GIXRF (grazing-incidence X-ray fluorescence)[31–34]. When the incident angle is below the critical angle for the total-external reflection of the topmost surface or any buried interface, the evanescence wave is created, and a standing wave is also generated above the interface. This is known as the total-external-reflection X-ray standing wave (TER-XSW). In this work, we take advantage of this effect by setting the incident angles less than the critical angle of the Pd substrate so that the standing wave effect dominates and thus the enhanced and spatially modulated E-field above the substrate can be used as a highly sensitive probe for the nanostructures. In contrast, in the grazing-exit configuration, GEXRF (grazing-exit X-ray fluorescence) that is often used for low-level chemical impurity detection[35–38], the external beam often impinges normal to the sample,

eliminating the need for beam collimation and sample surface alignment. With efficient area pixel-array detectors, the fluorescence holographical cones can be quickly measured without the need of scanning the detector position and still give sufficient angular resolution near the grazing-exit angles. Therefore, for in situ and time-resolved measurements, grazing-exit is more desired than grazing-incidence configuration. However, unlike the grazing-incidence configuration, the very large elastic scattering background arising from the normal or high incidence angle in the grazing-exit configuration[39] often complicates the data interpretation and harms the reconstruction quality. In contrast, these difficulties are readily overcome in the mixed-mode XWFH with simultaneous grazing-incidence and grazing-exit angles. In addition, while GIXRF and GEXRF have been occasionally used on the same sample[40–42], the measurements were carried independently rather than simultaneously like in XWFH. The additional information and constraints provided by the X-ray standing waves generated at simultaneous grazing-incidence and exit angles in the XWFH configuration can greatly facilitate the reconstruction speed and the resolution of the depth profiling.

In XWFH, the angular dependence of the fluorescence hologram is a result of interference of the fluorescence within the waveguide, giving clues of the electron density profile of the waveguide. Although a complete reconstruction of an XWFH hologram requires a dynamical theory that is more sophisticated than the kinematical theory, it takes into account multiple scattering, extinction, self-interference, etc. Such phenomena are not readily handled with conventional XFH reconstruction algorithms. In our dynamical holographical reconstruction for XWFH, the nanostructure of interest self-consistently enters the iterative computation of the electric field that subsequently illuminates the fluorescence substances to give out angular-resolved fluorescence holograms. In addition, the reconstructed density profile of the waveguide is required to converge to a solution conforming to X-ray standing wave conditions for all the engaged energies (elastic and all fluorescence energies). The use of multiple energies is equivalent to an expansion of the measurement dimension and serves as an additional constraint for the reconstruction. As a result, it removes the ambiguity of the solution uniqueness that is encountered in many inverse problems. In addition, the convergence speed of the algorithm and the reconstruction accuracy are improved. In practice, this is analogous to the multiple-energy X-ray holography in solving local atomic environments in crystals, where image distortions of a single-energy hologram such as twin images can be effectively suppressed[4]. An additional benefit of multiple emission energies in XWFH is that they provide an automatic self-calibration of the exit angle, which mitigates the challenge of angular calibration in conventional grazing-exit configuration[38]. For instance, the rising edge of the very first peak in the hologram corresponds to the critical angle of the polymer film (Fig. 3c, h) and its position is inversely related to the emission energy. We, therefore, can easily infer the absolute exit angle as well as the sample-detector distance from different rising edges arising from multiple energies. Besides, in many GIXRF and GEXRF experiments where the index of refraction or optical constant of the substrate has to be specified in advance and thus has great direct impacts on the accuracy of the analysis. However, it is less of a concern for XWFH, because many constraints imposed by the waveguide effect enable the simultaneous optimization of the index of refraction as well as the layer thickness and roughness of the substrate as long as the indices of refraction of at two layers are provided (here helium and silicon are fixed at tabulated values). Nevertheless, to reduce the model complexity, these substrate-related parameters can be pre-determined precisely with the high-

resolution reflectivity, a standard tool for X-ray mirror characterization, on the reference substrate before the film deposition. In summary, conventional fluorescence techniques such as GIXRF and GEXRF are two special variations of XWFH. Therefore, the full dynamical theory, as well as the efficient reconstruction method developed for the general scenario in XWFH, can be easily simplified for the data analysis in GIXRF and GEXRF.

The qualification of X-ray characterization techniques for in situ and time-resolved studies of elemental depth profile at the nanoscale is crucial to correlate material properties with the underlying chemical and physical properties. Combining many advantages of existent grazing-angle X-ray techniques, XWFH can serve as a powerful and high-resolution tool for quantitative thin-film and surface analysis when facilitated by the dynamical theory for waveguides and the novel reconstruction algorithm that we developed for XWFH. Although XWFH has proved in this work to deliver superior performance as compared to GISAXS and reflectivity, it is a complement to those conventional elastic scattering techniques and should be applied depending on the availability of experiment conditions and specific scientific problems to address. On the other hand, XWFH can be implemented as a scan-free technique, hence it is suitable for emergent systems that require in situ and time-resolving capabilities, as have been demonstrated in the present study. XWFH is also a flexible and non-contact technique and its capability can be extended and combined with other techniques. For example, one can vary the incidence angle to establish different modes for the incident X-ray standing waves in order for an even higher depth sensitivity. The incident energy can also be swept near the absorption edges of relevant substances[43], thus combining the X-ray absorption spectroscopy (XAS) analysis with XWFH to simultaneously obtain the chemical sensitivity of elements and the depth sensitivity of structures.

## Methods

**Materials and samples.** Samples are gold nanoparticle monolayers synthesized by thermal evaporation and sandwiched between two layers of poly (*tert*-butyl acrylate) (PtBA) of equal thickness of ~250 Å. PtBA of two molecular weights (19.6 kg/mol and 46.5 kg/mol) are used in this study, and both have a polydispersity index <1.2. The glass transition temperature $T_g$ of PtBA is ~49 °C. To prepare these thin films, silicon substrates are first coated with a chromium adhesive layer (~50 Å) and followed by a palladium mirror layer (~500 Å) in a thermal evaporation chamber. A PtBA layer is then coated onto the Pd mirror by spin-casting from a butanol solution. An ultrathin gold layer of a nominal thickness of ~6 Å is then deposited by thermal evaporation onto the PtBA layer. Gold nanoparticles form and grow spontaneously as a result of much weaker polymer-metal interaction than that between the metal atoms, and their sizes are typically around 2 nm[29,31,44]. A second PtBA layer is spun-cast onto a spare silicon substrate. It is then floated onto a water surface and picked up onto the top of the gold nanoparticle monolayer to create a sandwiched sample. Three types of samples are made by using different molecular weight for the supporting and capping PtBA layers. For notational convenience, they are denoted by two letters starting with the supporting followed by the capping layers: LL, HH, and LH, where L and H stand for low and high molecular weights, respectively. All samples are finally capped with a polystyrene (PS) layer (molecular weight of 131 kg/mol, polydispersity index of 1.08, thickness of ~250 Å, and $T_g$ of ~100 °C) to prevent dewetting upon thermal annealing.

**Experimental setup.** The experiments were performed at Sector 7-ID-C at the Advanced Photon Source, Argonne National Laboratory. The incident X-rays of energy 12.11 keV and bandwidth $\Delta E/E = 10^{-4}$ were collimated with two slit pairs so that the beam size was 400 μm × 60 μm (H × V) with a flux of $6 \times 10^{10}$ photons per second on the sample. Samples were loaded onto a thermal stage inside a sealed helium chamber in order to minimize the radiation damage and air scattering background. The chamber was mounted on a six-circle diffractometer with xyz translational stages. To further control the radiation damage, the sample was horizontally translated to a fresh region after one-minute exposure (far less than the sustainable lifetime estimated on similar test samples by monitoring the change of reflectivity and GISAXS). GISAXS and XWFH patterns were simultaneously collected with two Pilatus (Dectris, Inc.) single-photon counting pixel array detectors (pixel dimension of 0.172 × 0.172 mm²). The GISAXS detector was 996

mm away from the sample in the forward scattering direction, and the XWFH detector was 3180 mm at the right in-plane angle in order to minimize the elastic scattering background. In order to enhance the XWFH and GISAXS signals and meanwhile to further suppress the background contributions from the substrate, the incident angle was chosen to be 0.125° so that it is less than any critical angles of the sample support layers (Si, Cr, and Pd) but still resides above the critical angle of the polymer film. The reflectivity was measured by simultaneously scanning the incident and exit angles and recording the intensity within an ROI (region of interest) on the GISAXS detector. To minimize the air absorption and background scattering, vacuum flight paths were installed between the sample chamber and the two Pilatus detectors. XWFH, GISAXS, and reflectivity were first measured at the ambient temperature on each sample. The temperature was then ramped to 60 °C in about 7 min for thermal annealing experiments. To achieve a required time-resolution, only the scan-free XWFH and GISAXS were recorded during the thermal annealing. Reflectivity data were collected again at the end of the annealing cycle. The solid angles of pixels on each detector were determined and taken into account during the data analysis as an angular resolution convolution to a window function. The quantum efficiency for each energy (the efficiency curve is provided by the manufacturer) cannot be corrected beforehand, because Pilatus is not an energy-resolving detector and does not have a sufficient energy resolution to efficiently separate photons of different energies. However, according to the dynamical scattering theory or XWFH, the energy resolution is transformed into the angular resolution; thus the quantum efficiency correction can be iteratively applied (as a multiplicative combination with the air path absorption correction because they both effectively behave as the attenuation effect from the detection efficiency point of view) to the calculated XWFH for each energy during the reconstruction.

**Electric field calculation.** Given X-ray energy, the transmitted and reflected wave amplitudes $T(\alpha, z)$ and $R(\alpha, z)$ within a film or waveguide can be calculated for every height $z$ and angle $\alpha$ (incident or exit) using Parratt's recursive method which gives the exact one-dimensional solution of the Helmholtz equation for the stationary wave-propagation form of the Maxwell equations[45,46]. The complex electric field is given by

$$E(\alpha, z, \lambda) = T(\alpha, z)e^{ik_z(\alpha,z)z} + R(\alpha, z)e^{-ik_z(\alpha,z)z}, \quad (7)$$

where $k_z(\alpha, z)$ is the $z$-component of the wave vector for angle $\alpha$. $|E|^2$ is often called the electric field intensity (EFI). In general, the perturbation of a dense layer to the overall electron density, hence the electric field, cannot be ignored. A self-consistent multi-layer method is needed to obtain the correct electric field $E(z, \alpha, \lambda, \rho(z))$[30]. In this method, both $\rho(z)$ and wave vector $k_z(\alpha, z)$ are complex values so that the extinction and absorption effects automatically take place.

**Theory of GISAXS analysis.** The elastic forward scattering is quantitatively modeled in the framework of distorted wave Born approximation (DWBA)[47], where the nanostructure of interest (e.g., gold nanoparticles) is viewed as a perturbation to the reference scattering potential produced by the supporting substrate and the embedding film. Due to the perturbation of the gold layer to the overall electron density, a multi-layer form of the DWBA must be employed[30]. For a two-dimensional superlattice, the structure factor of the monolayer $S(q_{\parallel})$ only depends on the in-plane scattering angle $2\theta$ or equivalently the in-plane wave vector transfer $q_{\parallel}$. Using the local monodisperse approximation, the GISAXS intensity is given by[30]

$$I_G(\alpha_i, \alpha_f, 2\theta) \propto |\Delta\rho|^2 S(q_{\parallel}) \left\langle \left| \sum_{m=1}^{4} \int dz \phi_{np}(z) D^m(z) F[q^m(z), r] e^{iq_z^m(z)z} \right|^2 \right\rangle, \quad (8)$$

where $\Delta\rho$ is the constant electron density contrast between gold and the embedding polymer. The form factor $F[q^m(z), r]$ describes the morphology of the nanoparticles, and it is approximated by a sphere model,

$$F[q^m(z), r] = 4\pi r^3 \frac{\sin[q^m(z)r] - q^m(z)r\cos[q^m(z)r]}{[q^m(z)r]^3}. \quad (9)$$

$D^m(z)$ relates to the electric field in terms of the transmitted and reflected wave amplitudes

$$D^1(z) = T(\alpha_f, z)T(\alpha_i, z), D^2(z) = R(\alpha_f, z)T(\alpha_i, z), \quad (10)$$

$$D^3(z) = T(\alpha_f, z)R(\alpha_i, z), D^4(z) = R(\alpha_f, z)R(\alpha_i, z). \quad (11)$$

$q^m(z)$ is viewed as the wave vector transfer and is defined as

$$q^1(z) = \left\{ q_z^1(z), q_{\parallel} \right\} = \left\{ k_z(\alpha_f, z) - k_z(\alpha_i, z), q_{\parallel} \right\}, \quad (12)$$

$$q^2(z) = \left\{ q_z^2(z), q_{\parallel} \right\} = \left\{ -k_z(\alpha_f, z) - k_z(\alpha_i, z), q_{\parallel} \right\}, \quad (13)$$

$$q^3(z) = \left\{ q_z^3(z), q_{\parallel} \right\} = \left\{ -q_z^2(z), q_{\parallel} \right\}, \quad (14)$$

$$q^4(z) = \left\{ q_z^4(z), q_{\parallel} \right\} = \left\{ -q_z^1(z), q_{\parallel} \right\}. \tag{15}$$

the angle brackets <...> in Eq. (8) represent the polydispersity convolution over the nanoparticle size distribution. The gold nanoparticle number density is modeled as a Gaussian

$$\phi_{np}(z) = \frac{N_{np}}{\sqrt{2\pi\sigma_{np}^2}} e^{-\frac{(z-h_{np})^2}{2\sigma_{np}^2}}, \tag{16}$$

where $h_{np}$ and $\sigma_{np}$ are mean height and standard deviation above the Pd mirror, and $N_{np}$ is the total number of nanoparticles. In order to compute the electron density profile, and hence the transmitted and reflected wave amplitudes $T$ and $R$, the nanoparticle number density $\phi_{np}(z)$ needs to be converted to the atomic number density $\phi_{Au}(z)$ with the nanoparticle model parameters such as radius and polydispersity. These morphological parameters are either previously known or kept floating for fitting in GISAXS analysis.

**Reflectivity analysis**. The reflectivity calculation is accomplished as a byproduct of the Parratt's recursive method for the electric field computation. Briefly, it is the squared modulus of the amplitude of the topmost reflected wave at the film surface[45], i.e., $I_R(\alpha) = |R(\alpha)|^2$.

## Data availability

The source data (XWFH, GISAXS, and reflectivity) supporting the findings of this study are available within the paper and its supplementary information files.

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

## Acknowledgements
This research used resources of the Advanced Photon Source, a U.S. Department of Energy (DOE) Office of Science User Facility operated for the DOE Office of Science by Argonne National Laboratory under Contract No. DE-AC02-06CH11357. ZJ was supported by the DOE Early Career Research Program. The project is partially supported by an Argonne Laboratory Directed Research and Development fund. We also thank Prof. Kenneth Shull's group at Northwestern University for supporting the project with sample preparations and helpful discussions.

## Author contributions
Z.J. and J.W. conceived the idea and designed the experiment. All authors, Z.J., J.W.S., D.A.W., and J.W., carried out the experiment. Z.J. analyzed the data. Z.J. and J.W. wrote the manuscript.

## Competing interests
The authors declare no competing interests.
