## [Peer Review File · Nature Communications]

Peer Review File - Reviewers' comments first round:

Reviewer #1 (Remarks to the Author):

The manuscript of Z. Jiang et al. on the 'Reconstruction of Evolving Nanostructures in Ultrathin Films with X-ray Waveguide Fluorescence Holography' is a very valuable contribution to the related research field of non-destructive nanoanalytics for several reasons:

1. The qualification of x-ray characterisation techniques for in-situ and time-resolved studies of elemental depth profile at the nanoscale is crucial to correlate material properties with the underlying chemical and physical properties.
2. The simultaneous use of independent techniques such as the mixed-mode XWFH and GISAXS allow for the identification of potential artefacts like the slight broadening of depth profiles by GISAXS
3. It is necessary to clearly describe the need for a-priori Knowledge required by a method like GISAXS, i.e. the shape of nanoparticles and assumptions on the type of spatial distributions, when comparing two orthogonal methods.
4. The literature, advantages and disadvantages of related techniques such as GIXRF and GEXRF have been well discussed.

There are some minor points worthwhile to be corrected or added for a revised version of the manuscript:

- a. The impact of the energy dependence of the detection efficiency of the 2D fluorescence detector for the Au-L fluorescences lines used on the data evaluation.
- b. A qualitative (or perhaps even quantitative) comparison of the depth profiling capability of GIXRF, GEXRF and mixed-mode XWFH for buried nanolayer (or equivalent elemental distributions); also a reference to additional literature may serve for this Purpose
- c. an estimated budget for the uncertainties related to the measurands accessible by mixed-mode XWFH
- d. a few terms used in the manuscript are related to specific communities (e.g. fluorescence yield = fluorescence intensity or elastic energy = excitation energy) ; there is no need to change the entire manuscript but as readers may be from different fields it would be wise to use both terms when introducing them for the first time.

Reviewer #2 (Remarks to the Author):

The paper demonstrates X-ray waveguide fluorescence holography (XWFH) as a significant extension to X-ray fluorescence (XRF) holography by confining the XRF nano sources/scatterers within a thin film, which acts as a wave guide. They make use of the optical reciprocity theorem in the form of the evanescent-wave emission effect, for which the authors should reference Becker, Golovchenko, Patel PRL 50 (1983). The other clever trick performed in this study is to use both the Au La and Lb XRF signals for self-validation and for adding constraints to their analysis. Their work is validated by a comparison of the XWFH structural findings of the z-profile of the Au nanosphere layers to that found by the more established method of grazing incidence small angle X-ray scattering (GISAXS). To bring this up a notch it would help if the authors more clearly substantiated on page 13 the potential importance of this new technique over other competing techniques, such as GISAXS and conventional TER-XSW.

Reviewer #3 (Remarks to the Author):

The authors present a proof of principle experiment with a technique they label X-ray Waveguide Fluorescence Holography (XWFH). The principle is that a thin layer acts as a waveguide to form an X-ray standing wave field when illuminated at grazing incidence angles. By measuring X-ray fluorescence at grazing emission angles the authors analyze gold nanoparticle layers and their dynamics upon thermal annealing. They state, that the gold nanoparticles in this case act as the

fluorescence emitters as well as scattering centers and therefore dynamic scattering theory must be used for the reconstruction of the gold distribution.

The paper is well written and cites much of the relevant literature and the results are in large parts comprehensible. The problem with the work is that the novelty and significance of the paper is questionable.

As early as 1999, the enhancement of GEXRF due to sandwiching was published (*Spectrochimica Acta Part B: Atomic Spectroscopy* 54 1999 1881-1888) and the idea proposed that this could also be performed under GI conditions. In 2000, the same authors published GEXRF under two GI angles (*X-Ray Spectrom.* 2000; 29: 155–160). Recently, a few more examples of combining grazing incidence and emission subsequently for detecting X-ray fluorescence have been published (e.g. *J. Anal. At. Spectrom.*, 2012, 27, 1432-1438, *J. Anal. At. Spectrom.*, 2015, 30, 1086-1099, *J.SynchrotronRad.*(2015).22, 1419–1425). In all these publications, a sample as measured by the authors would be possible to measure and quantify.

Analytically, the authors do not compare or discuss thoroughly the uniqueness of their reconstruction compared to other published reconstruction algorithms. Additionally, the information on the simulations is very short and at least should have been elaborated in the SI. And last but not least, the authors neither show a significant scientific case as application, nor discuss what applications are only possible with their 'new' technique. In conclusion, while the work is interesting and scientifically sound, the lack of significance and novelty prevent a publication in nature communications.

Reviewer #1 (Remarks to the Author):

The manuscript of Z. Jiang et al. on the 'Reconstruction of Evolving Nanostructures in Ultrathin Films with X-ray Waveguide Fluorescence Holography' is a very valuable contribution to the related research field of non-destructive nanoanalytics for several reasons:

- 1. The qualification of x-ray characterisation techniques for in-situ and time-resolved studies of elemental depth profile at the nanoscale is crucial to correlate material properties with the underlying chemical and physical properties.*
- 2. The simultaneous use of independent techniques such as the mixed-mode XWFH and GISAXS allow for the identification of potential artefacts like the slight broadening of depth profiles by GISAXS*
- 3. It is necessary to clearly describe the need for a-priori Knowledge required by a method like GISAXS, i.e. the shape of nanoparticles and assumptions on the type of spatial distributions, when comparing two orthogonal methods.*
- 4. The literature, advantages and disadvantages of related techniques such as GIXRF and GEXRF have been well discussed.*

Response: We appreciate the reviewer for recognizing the merit of this work, and for acknowledging our contribution to the field.

There are some minor points worthwhile to be corrected or added for a revised version of the manuscript:
a. The impact of the energy dependence of the detection efficiency of the 2D fluorescence detector for the Au-L fluorescences lines used on the data evaluation.

Response: The detector (Pilatus) is not an energy-resolving detector. There is only one lower energy threshold. In theory, one is able to distinguish the L-alpha ($L_{\alpha 1}=9.713\text{keV}$, and $L_{\alpha 2}=9.628\text{keV}$) from L-beta (11.585keV) by two exposures with the energy thresholds set first between the two energies and then below the L-alpha. This, however, is impractical for this type of detector because the photon counts are unreliable if the energy threshold is far off the half of the energy of interest due to the charge spread and split across neighboring pixels. We, therefore, set the energy threshold at $\sim 5\text{keV}$ according to the manufacturer's recommendation and collected both L-alpha and L-beta in a single exposure. In fact, single exposures benefit time-resolved experiments. In addition, the coexistence of both energies imposes additional constraints to help the reconstruction algorithm to converge to a unique solution, which is demonstrated by the analysis (as also pointed out by reviewer 2). The quantum efficiencies of the detector for the two energies are from the detector's spec sheet. Since we cannot distinguish the two energies beforehand, the efficiency corrections were taken into account dynamically during the reconstruction where the efficiencies were applied to the predicted XWFH for each energy. We added these details to the Experiment section in Methods.

b. A qualitative (or perhaps even quantitative) comparison of the depth profiling capability of GIXRF, GEXRF and mixed-mode XWFH for buried nanolayer (or equivalent elemental distributions); also a reference to additional literature may serve for this Purpose

Response: As we discussed at the end of the paper, GIXRF and GEXRF are the two special cases of XWFH. As both reviewers 1 and 3 suggested, we add more relevant literature. The dynamical theory we developed here can also be simplified to analyze either GIXRF or GEXRF. In fact, we have performed the GIXRF on similar sandwiched sampled; and this work had been published as [Lee, D. R. et al.

Perturbation to the resonance modes by gold nanoparticles in a thin-film-based x-ray waveguide. Applied Physics Letters 88, 153101 (2006)], and it was already cited in the initial submission [Lee2006]. We now demonstrate the more general scenario with XWFH in this work. By applying the dynamical theory for the standing wave to the effects simultaneous from both grazing-incidence and exit (GIXRF and GEXRF), XWFH provides greater sensitivity and reconstruction confidence. In contrast, in either GIXRF or GEXRF alone, one would expect the quality of data analysis to be less satisfactory as shown in several references that were mentioned by Reviewer #3.

c. an estimated budget for the uncertainties related to the measurands accessible by mixed-mode XWFH

Response: To address this comment (also a concern shared by Reviewer #3 and possibly the readers of this article in general as well), we moved the Method Section “XWFH reconstruction algorithm” to the main text, and expanded it with more details about the Bayesian-inference (Hamiltonian Markov Chain Monte Carlo) based model-independent reconstruction algorithm that we developed for this work. Another general-purpose manuscript is under preparation for the algorithm itself for general experimental data analysis. For the completeness of the current work, we briefly described the method in the main text and included an introduction and the data analysis procedure in the Supplementary. We also evaluated the performance in terms of convergence and profile confidence/uncertainty. To make Figures 3 and 4 and the main text concise, we added this additional information in the Supplementary. In summary, both the intrinsic precision of X-ray standing wave and the efficient reconstruction algorithm leads to a fast and stable convergence to a unique solution with a very narrow uncertainty band for the spatial profile of the fluorescing elements. For example, the convergence of the reconstruction reached stationary in only a few iterations even with a very broader gold atom number distribution as the start guess value (Supplementary Figure S4 and S5).

d. a few terms used in the manuscript are related to specific communities (e.g. fluorescence yield = fluorescence intensity or elastic energy = excitation energy) ; there is no need to change the entire manuscript but as readers may be from different fields it would be wise to use both terms when introducing them for the first time.

Response: We accept these suggestions. Declarations have been made the first places these terms appear.

Reviewer #2 (Remarks to the Author):

The paper demonstrates X-ray waveguide fluorescence holography (XWFH) as a significant extension to X-ray fluorescence (XRF) holography by confining the XRF nano sources/scatterers within a thin film, which acts as a wave guide. They make use of the optical reciprocity theorem in the form of the evanescent-wave emission effect, for which the authors should reference Becker, Golovchenko, Patel PRL 50 (1983). The other clever trick performed in this study is to use both the Au La and Lb XRF signals for self-validation and for adding constraints to their analysis. Their work is validated by a comparison of the XWFH structural findings of the z-profile of the Au nanosphere layers to that found by the more established method of grazing incidence small angle X-ray scattering (GISAXS). To bring this up a notch it would help if the authors more clearly substantiated on page 13 the potential importance of this new technique over other competing techniques, such as GISAXS and conventional TER-XSW.

Response: We appreciate the reviewer for acknowledging the novelty part of our work. We also thank the reviewer for reminding us of the omission of the pioneering work by Becker et al. in the field; it is now included in the reference. In addition, to address the concerns on the completeness in the discussion part, we added a few sentences to review compare XWFH to other surface and thin-film based characterization techniques such as GISAXS, reflectivity, and TER-XSW.

Reviewer #3 (Remarks to the Author):

The authors present a proof of principle experiment with a technique they label X-ray Waveguide Fluorescence Holography (XWFH). The principle is that a thin layer acts as a waveguide to form an X-ray standing wave field when illuminated at grazing incidence angles. By measuring X-ray fluorescence at grazing emission angles the authors analyze gold nanoparticle layers and their dynamics upon thermal annealing. They state, that the gold nanoparticles in this case act as the fluorescence emitters as well as scattering centers and therefore dynamic scattering theory must be used for the reconstruction of the gold distribution.

The paper is well written and cites much of the relevant literature and the results are in large parts comprehensible. The problem with the work is that the novelty and significance of the paper is questionable.

Response: First we thank the reviewer for acknowledging the comprehensibility of our paper. But we disagree with the reviewer on the comments of the novelty and significance of our work. In the paper, we developed a new technique (XWFH) to extract the nanostructures in thin films. This technique is more comprehensive than similar techniques the reviewer mentioned such as GEXRF, GIXRF, etc. As we discussed in the manuscript, these existing techniques are special cases of our XWFH. In addition, performing XWFH *simultaneously* at both grazing-incident and grazing-exit angles and *simultaneously* analyzing multiple standing waves arising from multiple fluorescence signals impose very strong constraints on the reconstruction, solving uniqueness problem that has been encountered in many fluorescence imaging experiments. This setup and capability have *never* been reported anywhere else. In addition, XWFH is a high-precision scan-free technique, which is very suitable for in-situ and time-resolved experiments. Lastly, we also developed a novel and generalized inference-based model-independent reconstruction algorithm which facilitates fast convergence to the global minimum. This generalized nanostructure profiling algorithm has *never* been previously employed at all in any GEXRF, GIXRF or related experiments. While this new algorithm is the focus of our other general-purpose theory manuscript under preparation, for completeness, we added a brief introduction to this new reconstruction method and detailed data analysis procedures (See revision in the main text and Supplementary S3 and S4). We did not emphasize enough the novelty and significance of our technique in the original manuscript. However, the novelty and significance have been acknowledged by both other reviewers. Following Reviews 1 and 2 suggestions, we added a few statements in the conclusion of the revision. We hope this response will convince the reviewer that the work is novel and significant.

As early as 1999, the enhancement of GEXRF due to sandwiching was published (Spectrochimica Acta Part B: Atomic Spectroscopy 54 1999 1881-1888) and the idea proposed that this could also be performed under GI conditions. In 2000, the same authors published GEXRF under two GI angles (X-Ray Spectrom. 2000; 29: 155–160). Recently, a few more examples of combining grazing incidence and emission subsequently for detecting X-ray fluorescence have been published (e.g. J. Anal. At. Spectrom., 2012, 27, 1432-1438, J. Anal. At. Spectrom., 2015, 30, 1086-1099, J.SynchrotronRad.(2015).22, 1419–

1425). In all these publications, a sample as measured by the authors would be possible to measure and quantify.

Response: We thank the reviewer for pointing out the previous work. In fact, we are aware of most of the references and cited some of the most relevant ones. Here we took the opportunity to discuss the strength and weaknesses of the work mentioned in the reviewer's comments in regard to our work.

- GEXRF paper [Spectrochimica Acta Part B: Atomic Spectroscopy 54 1999 1881-1888] observed the enhancement effect in thin sandwiched films. However, it only dealt with exit angles, and no grazing-incidence effect was involved. The waveguide effect was observed, but quantitative data analysis was lacking, and the simulations were also visually far off from the experiment data.
- Paper (X-Ray Spectrom. 2000; 29: 155–160) performed both grazing-incident and exit angle measurements on a single supported fluorescing layer. However, *no standing wave or waveguide effect* (which is the important aspect in our XWFH technique) was involved nor discussed. Second, only single fluorescence energy for each element was analyzed. Third, some of the fitting results clearly showed significant deviations from experiment data especially near and above the critical angle which is most sensitive to the element distribution. For example in Figure 4f. the fitting to the data, especially Ni is very far off from satisfactory, as compared to our data analysis in our work. The issue there is likely due to the inadequate model and analysis method.

- Paper (J. Anal. At. Spectrom., 2012, 27, 1432-1438) use GEXRF and GIXRF to study the implanted aluminum layer in silicon. The authors did *not* simultaneously combine GEXRF and GIXRF into a *unified* technique, although the title is somehow misleading. In the paper, these two techniques were performed *independently* on the same sample. This is the major difference from our XWFH, where we kept emphasizing the feature of combining grazing-incidence and exit *simultaneously* in both measurement and reconstruction. Second, no clear standing wave took effect nor was it discussed. Third, the fitting quality was far from satisfying, as can be seen from the only GEXRF and GIXRF figure in the paper (copied here).

- Similar to the above paper, Paper (J. Anal. At. Spectrom., 2015 30, 1086-1099) did depth profiling of ion implantations in substrate. Although both GEXRF and GIXRF were used, they were independent measurements, instead of a unified technique like our XWFH. In addition, data modeling did not strictly take into account the dynamical effect.
- Paper (J.SynchrotronRad.(2015).22, 1419–1425) is the only paper that explicitly acknowledged the standing wave effect. However, again GIXRF and GEXRF were performed *independently*. The advantages of constraints on the profile reconstruction imposed by simultaneous GIXRF and GEXRF were ignored. In addition, the fitting quality was far from satisfaction as compared to our XWFH reconstruction, for example, in figure 5:

Listed below is a summary of the papers Review 3 mentioned, as well as novelties and advantages of our technique:

- Data modeling: There has been no details in those papers about theoretical models and data analysis procedures.
- Quality of analysis: The data fitting and simulations in those papers were visually off from experiment data.
- The generality of technique: GEXRF and GIXRF were performed independently in those papers, or the advantage of combining both techniques for data analysis was ignored. We have previously demonstrated GEXRF by scanning the incidence angle on similar layered samples. This work had been published as [Lee, D. R. et al. Perturbation to the resonance modes by gold nanoparticles in a thin-film-based x-ray waveguide. Applied Physics Letters 88, 153101 (2006)], and it was already cited in the initial submission [Lee2006]. We have stated in the present paper that GEXRF and GIXRF are two specific cases of the more general XWFH; thus we do not feel the necessity of repeating those special cases here, as we have already demonstrated the more general scenario (XWFH).
- Solution uniqueness: Those papers did not make use of the multiple energy strategy to improve reconstruction performance and accuracy. In contrast, we have shown that the use of multiple

energies is equivalent to the addition of another dimension in the measurement space, greatly improves the convergence and removes the ambiguity of the solution uniqueness in many inverse problems. The benefit of multiple energies has been acknowledged by Reviewer 2 and is demonstrated as the fast convergence and the stable solution in Supplementary Section S4.

- Applicable systems: Only inorganic systems were measured in those papers. In contrast, our technique, in general, is applicable to all films and surfaces regardless of materials: inorganic, organic (soft matter, polymer etc) or their composite films, for example, inorganic nanoparticles sandwiched in organic polymer layers in our paper. The radiation damage is particularly a problem for organic samples. Clearly, the scan-free version of the XWFH has a benefit by reducing the exposure time and thus preserving the samples.
- In-situ and time-resolved capability: The experiments described in those papers were all ex-situ experiments. In contrast, the scan-free version of the XWFH has been demonstrated to meet the in-situ time-resolved requirements for the study of the diffusion kinetics of nanoparticles. This type of capability has not been reported in any of the papers the reviewer mentioned.
- Dynamical theory and reconstruction algorithm: Our major concern on the data analysis in those papers is that the dynamical effect at interfaces and within layers has not been fully or properly accounted for, which partially explains the not-so-great simulation or fit to the data. Here, we developed the full dynamical theory as well as a novel and efficient reconstruction algorithm. We added more discussions about the relation of XWFH to GEXRF and GIXRF in the discussion section and emphasized the point that they are special cases of XWFH. The dynamical theory we developed can be easily simplified for analyzing data taken with those conventional techniques.

In summary, we believe our paper significantly distinguishes from all previous papers including the papers the reviewer listed.

Analytically, the authors do not compare or discuss thoroughly the uniqueness of their reconstruction compared to other published reconstruction algorithms. Additionally, the information on the simulations is very short and at least should have been elaborated in the SI.

Response: We thank the reviewer for showing great interest in the reconstruction method we used. It is not practical to compare to other published reconstruction algorithms we are aware of so far. Given the reasons above and already expressed in the original submission, the existing simulation or reconstruction algorithms simply do *not* work here because they did not rigorously consider the dynamical effect due to standing wave and waveguiding phenomena. As far as we know, our reconstruction method is the only and the accurate one that works for multilayered samples in the presence of dynamical effect and multiply fluorescence lines. We now emphasized these points in several places such as the XWFH Principle section and Discussion section.

To give more information on the XWFH reconstruction, which also was suggested by reviewer 1, we moved the reconstruction section from Method to the main text and significantly expanded it with the details about the algorithm and data reconstruction procedure. Now readers can refer to the revised manuscript and Supplementary for computational details.

And last but not least, the authors neither show a significant scientific case as application, nor discuss what applications are only possible with their 'new' technique. In conclusion, while the work is

interesting and scientifically sound, the lack of significance and novelty prevent a publication in nature communications.

Response: We respectfully disagree with the reviewer. In this paper, we aim to provide an innovative and high-precision methodology as well as a new reconstruction algorithm to reconstruct nanostructure profiles in thin films, rather than solely focusing on solving a specific application-related scientific problem. We believe the methodology presented here stands by itself in terms of novelty and significance. As we responded earlier, our XWFH is a generalization of GEXRF and GIXRF that the reviewer mentioned. All applications that those conventional techniques can be applied to our XWFH. In particular, the scan-free XWFH has proved to an in-situ and time-resolved technique that can be applied to fast and especially emergent systems. In addition, the dynamical effect has been largely ignored in the data interpretation of those conventional methods. The dynamical theory we developed here as well as the novel model-independent reconstruction algorithm can be simplified for either GE- or GI-XRF data analyses. In summary, we have demonstrated uniqueness and novelty on both the technique and reconstruction algorithm. We believe the community deserves to be aware of the availability of this new method and readily apply it either in experiments or in data analysis to address their specific scientific problems.

Reviewers' comments second round:

Reviewer #1 (Remarks to the Author):

The revised version of the manuscript of Z. Jiang et al. on the 'Reconstruction of Evolving Nanostructures in Ultrathin Films with X-ray Waveguide Fluorescence Holography' contains valuable updates on the issues raised with respect to the original submission:

a. The paragraph regarding the Pilatus detection efficiency added to the Experiment section in Methods is fine. One may add an estimate of the detector's spec uncertainty contribution to the overall uncertainty of the XWFH results.

b. The uncertainties of GIXRF and GEXRF depth profiling are strongly dependent on the uncertainties of the optical constants used for the calculation of the X-ray Standing Wave (XSW) field intensity distribution in Parratt's recursion formulation. For more pronounced spatial distributions of the elemental mass depositions than is the case in stratified layers, one has to use 2D- or 3D-Maxwell equation solvers to determine the correct XSW distribution in order to preserve the uncertainty of elemental depth profiles in the few percent range. However, it remains unclear to which extent the uncertainty of the XWFH depth profiles depends on the uncertainties of the optical constants employed and to which extent the dynamical approach used here reflects a 2D or 3D Maxwell equation solution.

c. Although the convergence of the XWFH approach is impressive, it remains somewhat unclear which uncertainty is associated with the depth profiles derived. Only figure S6 indicates 95 % confidence intervals associated with the HMCDC calculation results. In view of the authors' statement to b. that in either GIXRF or GEXRF one would expect the data analysis to be less satisfactory, it would be worthwhile to comment on this issue. In particular, several GIXRF or GEXRF studies have been validated by independent methods.

d. The use of alternate terms for yields and energies is acknowledged.

After a minor revision regarding the points mentioned above the manuscript may be accepted.

Reviewer #2 (Remarks to the Author):

The paper demonstrates X-ray waveguide fluorescence holography (XWFH) as a significant extension to X-ray fluorescence (XRF) holography by confining the XRF nano sources/scatterers within a thin film, which acts as a wave guide. This novel method makes use of the optical reciprocity theorem in the form of the evanescent-wave emission effect. The other clever trick is the use of both the Au La and Lb XRF signals for self-validation and for adding constraints to the analysis. The work is further validated by a comparison of the XWFH structural findings of the z-profile of the Au nanosphere layers to that found by the more established method of grazing incidence small angle X-ray scattering (GISAXS).

Reviewer #3 (Remarks to the Author):

The authors did a thorough job in replying to all comments and convincing me. The novelty and significance of the work has been highlighted and with the inclusion of the reconstruction details in the main text, the work is acceptable for publication.

Reply to Reviewer #1

The revised version of the manuscript of Z. Jiang et al. on the 'Reconstruction of Evolving Nanostructures in Ultrathin Films with X-ray Waveguide Fluorescence Holography' contains valuable updates on the issues raised with respect to the original submission:

a. The paragraph regarding the Pilatus detection efficiency added to the Experiment section in Methods is fine. One may add an estimate of the detector's spec uncertainty contribution to the overall uncertainty of the XWFH results.

b. The uncertainties of GIXRF and GEXRF depth profiling are strongly dependent on the uncertainties of the optical constants used for the calculation of the X-ray Standing Wave (XSW) field intensity distribution in Parratt's recursion formulation. For more pronounced spatial distributions of the elemental mass depositions than is the case in stratified layers, one has to use 2D- or 3D-Maxwell equation solvers to determine the correct XSW distribution in order to preserve the uncertainty of elemental depth profiles in the few percent range.

However, it remains unclear to which extent the uncertainty of the XWFH depth profiles depends on the uncertainties of the optical constants employed and to which extent the dynamical approach used here reflects a 2D or 3D Maxwell equation solution.

c. Although the convergence of the XWFH approach is impressive, it remains somewhat unclear which uncertainty is associated with the depth profiles derived. Only figure S6 indicates 95 % confidence intervals associated with the HMCMC calculation results. In view of the authors' statement to b. that in either GIXRF or GEXRF one would expect the data analysis to be less satisfactory, it would be worthwhile to comment on this issue. In particular, several GIXRF or GEXRF studies have been validated by independent methods.

Response: We appreciate the reviewer for acknowledging our efforts of improving the paper in the last revision. We feel that most of the reviewer's concerns about the uncertainty analysis can be addressed by clarifying the statistical and physical meanings of *uncertainty*, which is often misconceived in optimizations with conventional cost-minimization methods such as least-square fittings. All model parameters in the Bayesian-inference point of view are statistical variables. This means one does *not* optimize a problem by looking for a single set of parameters as the final solution; instead, the joint probability distribution of all the parameters that give a reasonable model approximation to experiment data is the goal. Given this distribution, we can evaluate the statistics (mean, mode, percentiles, variance, etc.) of the parameters as well as the statistics of variables or functions that are derived from these parameters such as model predictions. Following the conventions in statistics, the uncertainty or confidence in this paper is defined as the 95% confidence interval (between 2.5% and 97.5% percentiles). This distribution, however, does not take simple analytical expression for highly nonlinear and high-dimensional problems such as in many ill-posed inversion problems in the field of X-ray and neutron scattering, imaging or spectroscopy. Therefore, MCMC-based methods become popular recently and they draw a large number of samples from this distribution to numerically represent the joint distribution of the parameters and the probability distribution of the model predictions. Given these definitions, the replies to the reviewer's concerns are as follows:

(a) While the details statistical interpretation and uncertainty analysis of the MCMC result will be thoroughly discussed in our other HMCDC theory paper for inversion problems to be submitted for review, we feel it would be useful to discuss some guidelines for these uncertainties analysis here. Briefly speaking, in conventional cost-minimization methods (such as least-square minimization, etc.), the uncertainty of an individual parameter is often used to make evaluations of the dependence of the resulting uncertainty on the parameters. This isolated interpretation for the uncertainty is, however, somehow invalid or biased in the presence of parameter correlation (either physically correlated by causality, or statically correlated due to high nonlinearity of the model). In this work, the detector efficiency, if treated as an independent parameter, is fully correlated with the air path absorption (transmission), because both act effectively as energy-dependent attenuations. Therefore, it is invalid to evaluate the sole effect of the detector efficiency. As a result, one has to combine these two efficiencies as a single parameter during the HMCDC. To give another example, the thicknesses of the PtBA layer and PS layer are found to be highly anti-correlated from the covariance analysis of HMCDC-sampled thickness pairs of PtBA and PS layers. Besides, separate statistics analysis on the thicknesses of the two layers shows very low t-statistics, but the t-statistics of the sum of the two thicknesses (i.e. treat the entire polymer as a single layer) is extremely high. This means that isolated uncertainty analysis of the profile on one parameter is conditional on one or more other parameters, thus obscuring the meaning of uncertainty dependence analysis. Instead, the isolated uncertainty analysis is only valid on parameters that are completely uncorrelated with others; or more preferably the uncertainty dependence is analyzed on all parameters jointly, i.e. with respect to the joint probability distribution of the parameters.

(c) Another aspect of the uncertainty analysis, which is often misconstrued, is that the uncertainty of a measurement method only reflects the range of flexibility of the specific model applied for the analysis, and this uncertainty does not necessarily reflect the confidence with respect to the physical ground truth. According to the law of parsimony, the best practice is that one accepts a model that reasonably *well* (but *not the best* in terms of the χ^2 or sum of the residual squares as adopted in the least-square minimization) describes the measurement data and that is both physically validated as well as having the fewest number of parameters. For that purpose, cross-model assessment is required. To be specific, one needs to estimate parameters for all candidate models, and then compare them with model-selection criteria such as AIC, BIC, etc. Here, we apply the simple mean-square-residual (MSR) criteria, as shown in the inset of Figs 3c and 3h, to minimize the number of parameters for constructing the gold density profile. Unfortunately, we have not seen such a thorough cross-model assessment and within-model analysis in any previous GIXRF and GEXRF studies. However, we believe in the conclusions of the previous work because they have been validated experimentally with independent techniques. In this work, the high confidence about the profile with XWFH partially arises from the complete within-model and cross-model analysis, and this high-fidelity has also been experimentally validated with two independent and well-established techniques GISAXS and reflectivity.

(b) Optical constant uncertainty: While the substrate optical constant (also mass density given the chemical formula, or index of refraction in the present paper) is probably the primary factor that affects conventional GIXRF and GEXRF, etc. as they often need to be pre-determined and fixed before the analysis. Therefore, the accuracy of the substrate optical constant has a great direct impact on the accuracy of the profiling. In contrast, it is less of a concern in XWFH, because many constraints imposed by the waveguide effect enable the simultaneous optimization of the index of refraction as well as the

layer thickness and roughness of the substrate as long as the indices of refraction of at two reference layers are provided (here, the air and the silicon are fixed at tabulated values). Most important of all, the gold particles behave like a large perturbation to the optical constant of the film, causing z-dependence profiling of optical constant through the film so that modeling with a single static optical constant for the film is invalid in this work. Therefore, this optical constant due to the spread of gold (in the form of nominal gold layer thickness and the cubic spline coefficients), as well as other optical constants, are self-consistently included during reconstruction for the iterative calculations of the E-field. Detailed discussion on this self-consistent method, as well as the generation of XSW and its dependence on other parameters has been investigated thoroughly in our previous paper [reference #19 by Jiang et al., PRB 84, 75440 (2011)]. If one desires to minimize the number of parameters during the reconstruction, one can fix the optical constant and other parameters (thickness, roughness) of the substrate (for example, Si/Cr/Pd), which have been obtained from standard reflectivity analysis (as adopted in the high-resolution depth profiling method for X-ray mirrors and optics: see <http://www.cxro.lbl.gov>) on a reference substrate before depositing the polymers and gold layers. Overall, unlike GIXRF and GEXRF from a single substrate, where the optical constant is an independent parameter that may need to be fixed, it is less of concern in the presence of waveguide effect for the XWFH.

We do not think there is a need for the 2D- 3D Maxwell-equation solver. The XWFH in this paper focuses on depth profiling and does not take into account the in-plane structures. Therefore, only the z-dependent E-field is calculated. Parratt's method is derived from the Helmholtz equation which is the stationary wave-propagation form of the Maxwell equation for 1D, and the E-field calculated with the Parratt's method is the *exact* solution of the Maxwell equation. 2D or 3D Maxwell equation solver is required only when the film starts to develop 3D structures in the plane on a length scale such that there are a countable number of structural domains or clusters within the coherence lengths of the beam (typically several to a few hundred micrometers for ~12 keV at ~50-meter distance from the undulator synchrotron source) so that the ergodicity assumption for microscopic in-plane structural homogeneity breaks down. However, this is apparently not the case in the present study, considering the size of the nanoparticles.

To clarify the above points, we have added a few sentences (highlighted) in the text and a brief introduction of how to understand uncertainty in the Supplementary Information.

Reviewers' comments third round:

Reviewer #1 (Remarks to the Author):

I would like to thank the authors for their valuable and detailed response to my questions. The explanations given for the uncertainty analysis regarding statistical and physical constants related aspects are very much appreciated. The corresponding additions to the manuscript and supporting Information are fine as well.

I recommend the manuscript for publication without any further revision.

Author Response –

Reviewer #1 comment:

I would like to thank the authors for their valuable and detailed response to my questions. The explanations given for the uncertainty analysis regarding statistical and physical constants related aspects are very much appreciated. The corresponding additions to the manuscript and supporting Information are fine as well. I recommend the manuscript for publication without any further revision.

Response: We thank reviewer #1 for acknowledging our revision and the favorable recommendation for publication. All reviewers are greatly appreciated for recognizing our work and helping us improve our paper during the review process.